# Troponin T Assessment Allows for Identification of Mutation Carriers among Young Relatives of Patients with *LMNA*-Related Dilated Cardiomyopathy

**DOI:** 10.3390/jcm13113164

**Published:** 2024-05-28

**Authors:** Przemysław Chmielewski, Ilona Kowalik, Grażyna Truszkowska, Ewa Michalak, Joanna Ponińska, Agnieszka Sadowska, Katarzyna Kalin, Krzysztof Jaworski, Ilona Minota, Jolanta Krzysztoń-Russjan, Tomasz Zieliński, Rafał Płoski, Zofia Teresa Bilińska

**Affiliations:** 1Unit for Screening Studies in Inherited Cardiovascular Diseases, National Institute of Cardiology, 04-628 Warsaw, Poland; pchmielewski@ikard.pl (P.C.); emichalak@ikard.pl (E.M.);; 2Clinical Research Support Centre, National Institute of Cardiology, 04-628 Warsaw, Poland; 3Department of Medical Biology, National Institute of Cardiology, 04-628 Warsaw, Poland; 41st Department of Arrhythmia, National Institute of Cardiology, 04-628 Warsaw, Poland; 5Department of Coronary Artery Disease and Cardiac Rehabilitation, National Institute of Cardiology, 04-628 Warsaw, Poland; 6Department of Heart Failure and Transplantology, National Institute of Cardiology, 04-628 Warsaw, Poland

**Keywords:** *LMNA*, laminopathy, troponin, screening, hereditary DCM

## Abstract

**Background:** *LMNA*-related dilated cardiomyopathy (*LMNA*-DCM) caused by mutations in the lamin A/C gene (*LMNA*) is one of the most common forms of hereditary DCM. Due to the high risk of mutation transmission to offspring and the high incidence of ventricular arrhythmia and sudden death even before the onset of heart failure symptoms, it is very important to identify *LMNA*-mutation carriers. However, many relatives of *LMNA*-DCM patients do not report to specialized centers for clinical or genetic screening. Therefore, an easily available tool to identify at-risk subjects is needed. **Methods:** We compared two cohorts of young, asymptomatic relatives of DCM patients who reported for screening: 29 *LMNA* mutation carriers and 43 individuals from the control group. Receiver operating characteristic (ROC) curves for potential indicators of mutation carriership status were analyzed. **Results:** PR interval, N-terminal pro-B-type natriuretic peptide (NT-proBNP), and high-sensitivity cardiac troponin T (hscTnT) serum levels were higher in the *LMNA* mutation carrier cohort. Neither group differed significantly with regard to creatinine concentration or left ventricular ejection fraction. The best mutation carriership discriminator was hscTnT level with an optimal cut-off value at 5.5 ng/L, for which sensitivity and specificity were 86% and 93%, respectively. The median hscTnT level was 11.0 ng/L in *LMNA* mutation carriers vs. <3.0 ng/L in the control group, *p* < 0.001. **Conclusions:** Wherever access to genetic testing is limited, *LMNA* mutation carriership status can be assessed reliably using the hscTnT assay. Among young symptomless relatives of *LMNA*-DCM patients, a hscTnT level >5.5 ng/L strongly suggests mutation carriers.

## 1. Introduction

*LMNA*-related dilated cardiomyopathy (*LMNA*-DCM), caused by pathogenic variants in the lamin A/C gene (*LMNA*), is one of the most common forms of hereditary DCM, accounting for approximately 6% of DCM cases [1,2,3]. It is inherited in an autosomal dominant manner, so the risk of variant transmission to offspring is high (50%). *LMNA*-DCM is characterized by early onset, a frequent occurrence of cardiac conduction defects and arrhythmias, and an extremely unfavorable prognosis [4,5,6,7]. Therefore, relatives of *LMNA*-DCM patients should be subjected to clinical and genetic screening. Carriers of causative *LMNA* variants should be advised about lifestyle modifications, including career counselling and limiting physical activity, and should undergo periodic diagnostic tests, including standard ECG, echo, and ECG Holter recording [2,8]. Due to the high risk of ventricular arrhythmia and sudden cardiac death even before the onset of heart failure, it is very important to identify asymptomatic *LMNA*-mutation carriers. Some of them, despite normal left ventricular systolic function, may require protection with an implantable cardioverter-defibrillator. An arrhythmic risk calculator dedicated to *LMNA*-related cardiomyopathy is available [9].

Despite this, many relatives do not report for clinical or genetic screening. In many cases, psychological factors can be decisive. In other cases, access to specialist medical care may be hindered, and in many places, genetic testing, in particular, is still poorly accessible.

Therefore, an easily available tool to identify subjects at risk could be very useful. In a study published in 2020, we showed that elevated serum levels of cardiac troponin T are often found in young *LMNA* mutation carriers as the first abnormality, preceding conduction defects and arrhythmias [10]. In this study, we aimed to investigate whether the assessment of cardiac troponin T serum level using a high-sensitivity assay (hscTnT) helps to identify mutation carriers among young (<45 years of age), asymptomatic relatives of *LMNA*-DCM patients.

## 2. Materials and Methods

This study was conducted retrospectively and prospectively and included young (aged 18–45), asymptomatic or scantly symptomatic relatives of patients with the two most common forms of inherited DCM, *LMNA*-DCM and DCM associated with truncating variants of the titin gene (*TTN*), who reported to our unit for screening between 2013 and 2024. Previously known heart diseases, e.g., arrhythmia, as well as diseases that could affect the results of the tests, e.g., chronic kidney disease, were exclusion criteria. Non-specific and mild symptoms, such as heart palpitations or subjectively unsatisfactory physical performance, were acceptable if they did not constitute a reason to seek medical advice. All *LMNA* and *TTN* variants identified in the probands were pathogenic or likely pathogenic according to the American College of Medical Genetics and Genomics (ACMG) criteria (Table 1) [11]. All the participants underwent an interview, physical examination, 12-lead electrocardiography (ECG), echocardiography, and blood collection for genetic and biochemical testing. Genetic testing only included Sanger sequencing for the variant identified in the proband.

The study group consisted of asymptomatic carriers of a causative *LMNA* variant, and the control group of relatives in whom the presence of a causative *LMNA* or *TTN* variant was excluded. We compared selected parameters that could be potential indicators of *LMNA* mutation carrier status between the study and control groups. These included serum measurements of creatine kinase activity (CK), concentration of N-terminal prohormone B-type natriuretic peptide (NT-proBNP), and high-sensitivity cardiac troponin T (hscTnT), as well as PR interval measured on a standard ECG, and left-ventricular ejection fraction (LVEF) assessed with biplane Simpson’s method. The serum levels of NT-proBNP and hscTnT were measured by using the electrochemiluminescence immunoassays Elecsys proBNP II and Elecsys Troponin T hs STAT (both Roche, Mannheim, Germany), respectively.

All the results for the categorical variables were presented as numbers and percentages and, for continuous variables, as mean and standard deviation (SD) or median and quartiles (Q1:25th–Q2:75th percentiles). Fisher’s exact test was used for the comparison of categorical variables. The differences between continuous variables were tested by using the independent Student’s *t*-test (for two independent samples and for paired observation, appropriate, normally distributed data) or, in the case of skewed distribution, non-parametric Mann–Whitney U tests. 

A receiver operating characteristic curve (ROC) analysis was used to assess the cut-off point for the prediction of a variant carriership. The optimal cut-off was defined as the value with the maximal sum of sensitivity and specificity (Youden’s index).

All statistical analyses were performed using SAS 9.4 (Durham, NC, USA).

## 3. Results

The study group consisted of 29 asymptomatic *LMNA* mutation carriers from 15 families, and the control group was composed of 43 mutation-free relatives from 25 families of patients with *LMNA* and *TTN*-related DCM (9 and 16 families, respectively). *LMNA* mutation carriers were younger, and both groups did not differ significantly concerning LVEF (Table 2). The hscTnt level was below the detection threshold (<3.0 ng/L) in most subjects from the control group. In the *LMNA* mutation carriers’ group, it was usually still within the normal range but significantly higher (median 11 ng/L, *p* <0.001; normal range <14 ng/L). PR interval and NT-proBNP serum levels were also higher in the *LMNA* mutation carriers’ cohort.

Based on the C-statistics and Youden’s indices, the best *LMNA* mutation carriership indicator was the hscTnT level (Table 3).

The optimal hscTnT cut-off value for *LMNA* mutation detection was 5.5 ng/L, with a sensitivity of 86% and specificity of 93% (Figure 1). Other potential discriminators were characterized by low sensitivity (50–59%) at the optimal cut-off points. 

Of note, these findings were specific to *LMNA* mutation carriers. Asymptomatic carriers of DCM-causative *TTN* variants did not differ from the control group in terms of NT-proBNP and hscTnT levels (29 vs. 19 pg/mL, *p* = 0.20 and <3.0 vs. <3.0 ng/L, *p* = 0.81, respectively), despite significantly lower LVEF (Table 4).

## 4. Discussion

The principal new finding of our study is that the measurement of hscTnT serum concentration can be reliably used to determine young relatives of *LMNA*-DCM patients that may be carriers of the causative *LMNA* variant and that should therefore be subject to medical surveillance. We are far from saying that it can replace genetic testing; however, a hscTnT level > 5.5 ng/L has an excellent sensitivity and specificity of 86% and 93%, respectively, in detecting *LMNA* variant carrier status.

This finding may be treated as a “red flag” for several reasons. First, it can be used, whenever access to genetic testing is difficult, to emphasize the need for periodic screening. On the other hand, it could expedite genetic testing wherever it is feasible. Furthermore, hscTnT measurement may also be used when relatives do not report for screening due to procedural reasons (need to obtain referrals, waiting time, need to make several visits). The great advantage of troponin measurement is its widespread availability and low cost.

A prolonged PR interval as well as increased CK activity, reflecting skeletal muscle involvement, are often considered a marker of *LMNA*-DCM. Based on our analysis, hscTnT is a better discriminator of asymptomatic *LMNA* variant carrier status than those markers.

It may seem arguable that the hscTnT concentration values found in asymptomatic *LMNA* mutation carriers are usually still within normal limits. However, it should be noted that hscTnT levels are usually very low and often undetectable in young people. In a multicenter study, the mean hscTnT concentration was 4.0 ng/L in a cohort of 533 healthy individuals aged 20–71 [12]. The mean hscTnT levels were 2.7 and 2.6 ng/L in the case of healthy women aged 20–29 and 30–39, respectively, and 4.5 and 4.3 ng/L in the case of healthy men aged 20–29 and 30–39, respectively [12]. In the case of our study, the small study group did not allow us to conduct separate analyses for both genders.

The hscTnT values observed in the asymptomatic *LMNA* mutation carriers, although usually still normal, are substantially higher. This observation is consistent with the results of our previous study, in which we showed that elevated hscTnT serum levels (>14 ng/L) are often found in young *LMNA* mutation carriers as the first abnormality, preceding conduction defects and arrhythmias [8]. Nonetheless, this observation cannot be extrapolated to other forms of hereditary DCM. In our study on *TTN*-related DCM, elevated hscTnT levels could only be detected later, in the end-stage phase of the disease [13]. In asymptomatic *TTN* mutation carriers, we did not detect higher hscTnT levels compared to the control group during routine screening.

Cardiac troponin release can occur in many clinical situations, which may make the interpretation of test results more difficult [14,15]. It primarily occurs in many acute conditions; however, these are extremely unlikely in the context of screening a young and asymptomatic population. Other potential causes include chronic and often scantly symptomatic diseases, such as hypertrophic cardiomyopathy, valvular defects, or chronic renal failure, which should be considered during subsequent diagnostic workup [14,15,16]. Additionally, hscTnT levels may also increase after strenuous exercise [17].

False-positive troponin elevation should also be considered. It may result from the presence of heterophilic or human anti-animal antibodies, which may be present due to previous therapy with monoclonal antibodies, vaccinations, or even pet keeping. It has also been reported that false-positive results may be a consequence of the competitive interaction of diagnostic antibodies with fibrin clots, autoantibodies, or skeletal troponin molecules [15,18]. Nevertheless, in this study’s cohort, the *LMNA* variant carrier status was by far the most likely explanation for the higher hscTnT levels.

In *LMNA*-related DCM, an elevated hscTnT concentration could reflect subtle ongoing cardiomyocyte injury; however, the precise mechanisms of troponin leakage are poorly understood. It is believed that the nucleus acts as a mechanosensor through its connection to the cytoskeleton and extracellular matrix [19]. Abnormal lamins could lead to a substantial disruption of nuclear mechanobiological processes [19,20,21] and, in some situations, to obliteration of nuclear architecture [22]. The aftermath of prolonged cardiomyocyte injury is non-specific replacement fibrosis, as seen in endomyocardial biopsy or as late gadolinium enhancement in the CMR study [23,24,25]. Consequently, myocardial fibrosis is considered to be responsible for the development of electrical instability, leading to arrhythmia [6,26,27,28,29,30,31], and, over time, to mechanical impairment.

It would be worth investigating whether cardiac troponin I levels are also higher in *LMNA* mutation carriers. It should be noted that unlike hscTnT, which is measured by only one commercially available test, many different cardiac troponin I tests are available, including point-of-care devices.

It seems that the potential of serum biomarkers in the evaluation of patients with cardiomyopathies is not fully utilized. They are routinely used in diagnostics, and to a lesser extent in risk stratification [16,32,33]. Our study shows that their assessment may also provide additional information in the context of particular forms of hereditary cardiomyopathies [34].

## 5. Conclusions

Whenever access to genetic testing is limited, *LMNA* mutation carrier status can be reliably assessed using the hscTnT assay. Among young asymptomatic relatives of *LMNA*-DCM patients, a hscTnT level >5.5 ng/L strongly suggests mutation carriers. Relatives with higher hscTnT levels should be prioritized for genetic testing.

## Figures and Tables

**Figure 1 jcm-13-03164-f001:**
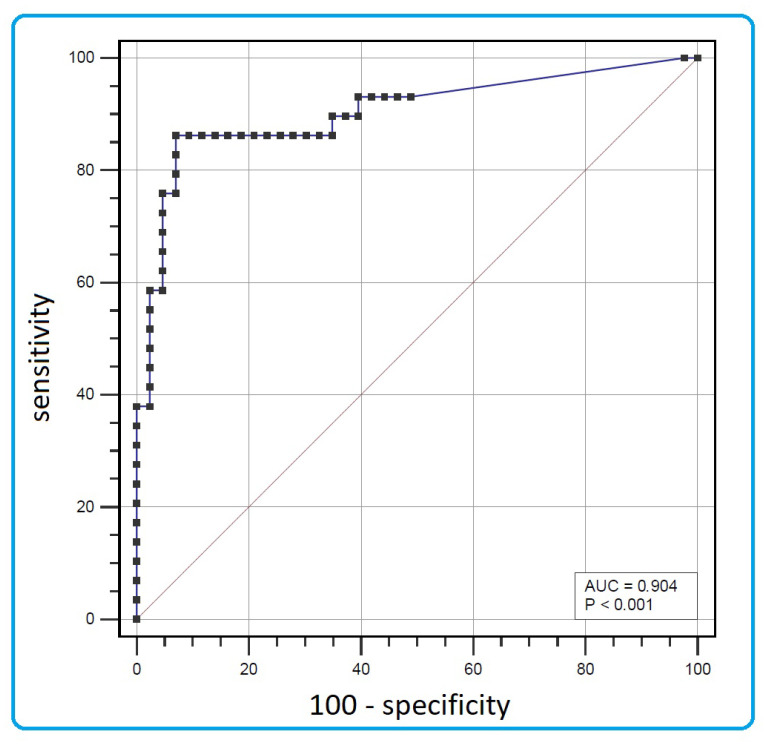
The receiver-operating characteristic curve for cardiac troponin T serum concentration as a discriminator of *LMNA* mutation carrier status.

**Table 1 jcm-13-03164-t001:** Genetic variants identified in the probands.

Gene	Location (hg 38)	NM_170707.4 (*LMNA*) NM_001267550.2 (*TTN*)	Protein	ACMG Classification
*LMNA*	1:156114934 C>T	c.16C>T	p.Gln6Ter	Pathogenic
*LMNA*	1:156115051 TA>T	c.134delA	p.Tyr45SerfsTer51	Pathogenic
*LMNA*	1:156115192 C>CT	c.276dupT	p.Asp93Ter	Pathogenic
*LMNA*	1:156134464 A>G	c.575A>G	p.Asp192Gly	Likely pathogenic
*LMNA*	1:156134491 A>AGGAG	c.607_608insGGAG	p.Glu203GlyfsTer12	Pathogenic
*LMNA*	1:156134901 C>T	c.736C>T	p.Gln246Ter	Pathogenic
*LMNA*	1:156136033 G>A	c.1069G>A	p.Asp357Asn	Likely pathogenic
*LMNA*	1:156136348 C>G	c.1292C>G	p.Ser431Ter	Pathogenic
*LMNA*	1:156136983 C>G	c.1443C>A	p.Tyr481Ter	Pathogenic
*LMNA*	1:156137144 G>GC	c.1526dupC	p.Thr510TyrfsTer42	Pathogenic
*LMNA*	1:156137173 C>T	c.1549C>T	p.Gln517Ter	Pathogenic
*LMNA*	1:156137666-C>G	c.1621C>G	p.Arg541Gly	Likely pathogenic
*LMNA*	1:156137666 C>T	c.1621C>T	p.Arg541Cys	Likely pathogenic
*TTN*	2:178800564 G>T	c.414C>A	p.Tyr138Ter	Pathogenic
*TTN*	2:178793462 G>T	c.1478C>A	p.Ser493Ter	Pathogenic
*TTN*	2:178632646 C>A	c.43360G>T	p.Glu14454Ter	Pathogenic
*TTN*	2:178630272 AT>A	c.44249del	p.Asn14750MetfsTer14	Pathogenic
*TTN*	2:178630240 C>T	c.44281+1G>A	n/a	Pathogenic
*TTN*	2:178612442 G>A	c.50083C>T	p.Arg16695Ter	Pathogenic
*TTN*	2:178607482 G>A	c.53206C>T	p.Arg17736Ter	Pathogenic
*TTN*	2:178598957 CCT>C	c.56751_56752del	p.Gly18918ValfsTer17	Likely pathogenic
*TTN*	2:178595585 G>A	c.57769C>T	p.Arg19257Ter	Pathogenic
*TTN*	2:178590527 C>CAT	c.61197_61198insAT	p.Gly20400MetfsTer6	Pathogenic
*TTN*	2:178589384 CAGTT>C	c.62337_62340del	p.Thr20780SerfsTer32	Pathogenic
*TTN*	2:178588700 G>A	c.63025C>T	p.Arg21009Ter	Pathogenic
*TTN*	2:178585271-A>T	c.64473T>A	p.Tyr21491Ter	Pathogenic
*TTN*	2:178578066-G>A	c.68449C>T	p.Arg22817Ter	Pathogenic
*TTN*	chr2-178575635 C>CA	c.70496dup	p.Leu23499PhefsTer3	Pathogenic
*TTN*	2:178572623 AT>A	c.73508del	p.Asn24503IlefsTer24	Pathogenic
*TTN*	2:178572397 T>TGTGG	c.73734_73735insCCAC	p.Lys24579ProfsTer11	Pathogenic
*TTN*	2:178567153 G>A	c.78979C>T	p.Arg26327Ter	Pathogenic
*TTN*	2:178565122 G>A	c.81010C>T	p.Gln27004Ter	Pathogenic
*TTN*	2:178561627 A>AT	c.84504dup	p.Ser28169IlefsTer12	Pathogenic
*TTN*	2:178557998 GC>G	c.87355delG	p.Ala29119LeufsTer17	Pathogenic
*TTN*	2:178557504 CT>C	c.87757del	p.Ser29253AlafsTer18	Likely pathogenic
*TTN*	2:178554642 AGT>A	c.88703_88704del	p.His29568LeufsTer7	Pathogenic
*TTN*	2:178549426 T>TG	c.92199dup	p.Asn30734GlnfsTer17	Pathogenic
*TTN*	2:178548460 G>A	c.93166C>T	p.Arg31056Ter	Pathogenic
*TTN*	2:178535388 G>A	c.101227C>T	p.Arg33743Ter	Pathogenic

Legend: ACMG, American College of Medical Genetics and Genomics.

**Table 2 jcm-13-03164-t002:** Comparison of asymptomatic *LMNA* mutation carriers and controls.

	*LMNA* Mutation Carriers *n* = 29	Control Group *n* = 43	*p*
Age at screening [years]	22 [19–28]	28 [21–37]	0.031
Male sex	15 (52%)	23 (53%)	0.883
Arterial hypertension	3 (10%)	1 (2%)	0.296
Other co-morbidities (DM, CKD, CAD)	0	0	1.00
Beta-blockers	4 (14%)	2 (5%)	0.212
ACE-I/ARB	1 (3%)	1 (2%)	1.00
Palpitations	3 (10%)	4 (9%)	1.00
Analyzed markers			
LVEF [%] n = 71	61 ± 5	60 ± 6	0.344
PR interval [ms]	154 [136–172]	135 [127–147]	0.025
Creatinine [mg/dL] n =65	0.80 ± 0.14	0.81 ± 0.14	0.767
NT-proBNP [pg/mL] n = 49	46 [15–89]	19 [7–34]	0.009
Creatine kinase [U/L] n = 70	143 [93–211]	110 [73–133]	0.067
hscTnT [ng/L]	11.0 [6.3–15.1]	<3.0 [<3.0–4.7]	<0.001

Number of subjects expressed as n (%). Continuous variables are shown as mean ± standard deviation or median and quartiles [Q1:25th–Q2:75th percentiles]. DM, diabetes mellitus; CKD, chronic kidney disease; CAD, coronary artery disease; ACE-I, angiotensin converting enzyme inhibitors; ARB, angiotensin receptor blockers; LVEF, left ventricular ejection fraction; NT-proBNP, N-terminal pro-B-type natriuretic peptide serum concentration; hscTnT, high-sensitivity cardiac troponin T serum level.

**Table 3 jcm-13-03164-t003:** Comparison of potential discriminators of *LMNA* mutation carrier status.

	Area under ROC Curve [95% CI]	*p*-Value Log-Rank	Optimal Cut-off	Sensitivity [95% CI]	Specificity [95% CI]	Youden’s Index	*p*-Value χ^2^ Test
hscTnT	0.90 [0.81–0.96]	<0.001	>5.5 ng/L	86% [74–99%]	93% [81–99%]	0.79	<0.001
PR interval	0.66 [0.54–0.77]	0.022	>147 ms	59% [39–76%]	77% [61–88%]	0.36	0.002
NT-proBNP	0.72 [0.58–0.84]	0.003	>48 pg/mL	50% [29–67%]	95% [76–100%]	0.45	<0.001

Legend: ROC, receiver operating characteristic; CI, confidence interval; hscTnT, high-sensitivity cardiac troponin T serum concentration; NT-proBNP, N-terminal pro-B-type natriuretic peptide serum concentration.

**Table 4 jcm-13-03164-t004:** Comparison of asymptomatic *TTN* mutation carriers and controls.

	*TTN* Mutation Carriers *n* = 34	Control Group *n* = 43	*p*
Age at screening [years]	27 [20–38]	28 [21–37]	0.996
Male sex	18 (53%)	23 (53%)	1.00
Arterial hypertension	3 (9%)	1 (2%)	0.316
Other co-morbidities (DM, CKD, CAD)	0	0	1.00
Beta-blockers	0	2 (5%)	0.500
ACE-I/ARB	1 (3%)	1 (2%)	1.00
Palpitations	0	4 (9%)	0.125
Analyzed markers			
LVEF [%] *n* = 76	55 ± 7	60 ± 6	0.001
PR interval [ms]	153 [133–164]	135 [127–147]	0.027
Creatinine [mg/dL] *n* = 68	0.77 ± 0.15	0.81 ± 0.14	0.189
NT-proBNP [pg/mL] *n* = 41	29 [22–59]	19 [7–34]	0.199
Creatine kinase [U/L] *n* = 73	109 [81–152]	110 [73–133]	0.590
hscTnT [ng/L]	<3.0 [<3.0–4.4]	<3.0 [<3.0–4.7]	0.810

Number of subjects is expressed as n (%). Continuous variables are shown as mean ± standard deviation or median and quartiles [Q1:25th–Q2:75th percentiles]. DM, diabetes mellitus; CKD, chronic kidney disease; CAD, coronary artery disease; ACE-I, angiotensin converting enzyme inhibitors; ARB, angiotensin receptor blockers; LVEF, left ventricular ejection fraction; NT-proBNP, N-terminal pro-B-type natriuretic peptide serum concentration; hscTnT, high-sensitivity cardiac troponin T serum level.

## Data Availability

The data presented in this study are available on request from the corresponding author due to legal reasons.

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
