# Peer review of "Troponin T Assessment Allows for Identification of Mutation Carriers among Young Relatives of Patients with LMNA-Related Dilated Cardiomyopathy"

_jcm, 2024, doi:10.3390/jcm13113164_

Round 1
Reviewer 1 Report
Comments and Suggestions for Authors
Congratulations to authors as they’ve given many hints to uncover patient with LMNA Carriers. The study is indeed interesting and well conducted; I just have some comments that I think they can improve it:
In order to empower the use of hscTnt, authors should discuss the inner biological variability of this biomarkers, taking into account potential false negatives or false positives.
If it is possible, I would suggest to include the tables in supplementary files into the main paper as they are such interesting
Moreover, in order to give a clinical context to the paper, I strongly suggest to add the prevalnce of LMNA DCM in young patients including this reference to their work “Inherited Arrhythmias in the Pediatric Population: An Updated Overview. Medicina (Kaunas). 2024 Jan 3;60(1):94. doi: 10.3390/medicina60010094. PMID: 38256355; PMCID: PMC10819657.”
Comments on the Quality of English Language
minor english revisions are needed
Author Response
Reviewer 1
Congratulations to authors as they’ve given many hints to uncover patient with LMNA Carriers. The study is indeed interesting and well conducted; I just have some comments that I think they can improve it:
Thank you for this opinion and for your suggestions.
In order to empower the use of hscTnt, authors should discuss the inner biological variability of this biomarkers, taking into account potential false negatives or false positives.
Thank you for this suggestion. We expanded the discussion to include these issues
If it is possible, I would suggest to include the tables in supplementary files into the main paper as they are such interesting.
The aim of our study was not to assess carriers of variants in TTN, but in LMNA, and the inclusion of families with TTN in the study arose from the need to enlarge the control group.
Our attempts to enlarge the study group by inviting researchers from other centers were unsuccessful (most responded that they did not have the necessary data).
Therefore, to enlarge the control group, we decided to include clinically and genetically negative patients from families with TTN-DCM.
As a side note to this work - which is why we included the table in the supplementary materials - we decided to also compare asymptomatic carriers of TTN variants - to highlight the differences in the clinical profiles of carriers depending on the genetic background of DCM.
Following your suggestion, we are moving both supplementary tables to the main body of the manuscript.
Thank you for highlighting this.
Moreover, in order to give a clinical context to the paper, I strongly suggest to add the prevalnce of LMNA DCM in young patients including this reference to their work “Inherited Arrhythmias in the Pediatric Population: An Updated Overview. Medicina (Kaunas). 2024 Jan 3;60(1):94. doi: 10.3390/medicina60010094. PMID: 38256355; PMCID: PMC10819657.”
Thank you for this suggestion. We expanded the Introduction section by adding appropriate formulations and references.
Reviewer 2 Report
Comments and Suggestions for Authors
I think this is a good paper with a useful result and written in a classical scientific way. I have some minor comments that can help to improve the manuscript, if the authors and the editor agree with them.
-Please, check the references. For example [7] does not support the statement in the sentence;
- Line 134: word 'Troponin' seems to be missing;
- Line 144: I would recommend changing 'is' to 'can be';
- Lines 146-147: Please consider rephrasing this sentence with more sure that your data is new and refusing to highlight the dedicated specialists;
Conclusion section: I would recommend adding selection for genetic testing as another benefit of troponin measurement.
Comments on the Quality of English Language
Editing of English language required.
Author Response
Reviewer 2
I think this is a good paper with a useful result and written in a classical scientific way. I have some minor comments that can help to improve the manuscript, if the authors and the editor agree with them.
Thank you for this favorable opinion and for your comments.
-Please, check the references. For example [7] does not support the statement in the sentence;
Thank you for noticing this. It has been corrected.
- Line 134: word 'Troponin' seems to be missing;
Thank you for noticing this. It has been corrected.
- Line 144: I would recommend changing 'is' to 'can be';
Done. Thank you for this suggestion.
- Lines 146-147: Please consider rephrasing this sentence with more sure that your data is new and refusing to highlight the dedicated specialists;
The paragraph has been rephrased. Thank you for this suggestion.
Conclusion section: I would recommend adding selection for genetic testing as another benefit of troponin measurement.
Thank you for this suggestion. An appropriate sentence has been added.